# Variability and Reliability of the Axivity AX6 Accelerometer in Technical and Human Motion Conditions

**DOI:** 10.3390/s25082480

**Published:** 2025-04-15

**Authors:** Marcos Echevarría-Polo, Pedro J. Marín, Esther Pueyo, Javier Ramos Maqueda, Nuria Garatachea

**Affiliations:** 1EXER-GENUD “Growth, Exercise, NUtrition and Development” Research Group, University of Zaragoza, 50009 Zaragoza, Spain; nugarata@unizar.es; 2Faculty of Health and Sports Sciences, University of Zaragoza, 22001 Huesca, Spain; 3CYMO Research Institute, 47140 Valladolid, Spain; pedrojm80@hotmail.com; 4BSICos Group, I3A, IIS Aragón, University of Zaragoza, 50018 Zaragoza, Spain; epueyo@unizar.es (E.P.); javierramosmaqueda7@gmail.com (J.R.M.); 5Centro de Investigación Biomédica en Red de Bioingeniería, Biomateriales y Nanomedicina, 50018 Zaragoza, Spain; 6Arrhythmias Unit, Cardiology Department, Hospital Clínico Universitario Lozano Blesa, 50009 Zaragoza, Spain; 7GIIS032 Research Group, Centro de Investigación Biomédica de Aragón, 50009 Zaragoza, Spain

**Keywords:** physical activity, accelerometry, activity monitor, reliability

## Abstract

This study aimed to evaluate the intra- and inter-instrument variability and reliability of the Axivity AX6 accelerometer under controlled technical conditions and human motion scenarios. In the first experiment, 12 accelerometers were affixed to a vibration platform and tested at four frequencies (2.2, 3.2, 6.5, and 9.4 Hz) along three axes to assess frequency- and axis-dependent variability. In the second experiment, four AX6 accelerometers were simultaneously placed on a subject’s wrist and tested under four human motion conditions (walking at 4 km·h^−1^ and 6 km·h^−1^ and running at 8 km·h^−1^ and 10 km·h^−1^). Results demonstrated low intra- and inter-instrument variability (CVintra: 3.3–4.5%; CVinter: 6.3–7.7%) with high reliability (ICC = 0.98). Similar results were observed in human motion conditions (CVintra: 5.3–8.8%; CVinter: 7.1–10.4%), with ICC values of 0.98 for combined devices, and 0.99 for each device individually. Despite statistically significant differences (*p* < 0.05) between devices in human motion all conditions, the variations remained below the minimal clinically significant difference threshold. These findings indicate that under technical conditions on a vibrating platform, and within the range of typical human accelerations, the Axivity AX6 is a reliable tool for measuring accelerations representative of physical activity. However, further research is necessary to validate its performance under free-living conditions.

## 1. Introduction

Activity monitors are wearable devices that are commonly used for monitoring free-living physical activity (PA) and even sleep parameters [1,2]. The popularity of these wearables has grown mainly due to their lower cost and greater accessibility and because their designs are more discreet and useful in their application for assessing PA [1]. In recent years, many commercial brands of accelerometers have been launched on the market. Axivity accelerometers have been widely used in research studies [3,4,5,6], with the UK Biobank standing out as one of the most significant and impactful [7]. The first commercially available device from Axivity, the AX3, was equipped with the ADXL345 triaxial acceleration sensor. With recent technological advancements, the company launched a more advanced model in 2019, the Axivity AX6. This model integrates the Bosch BMI160 acceleration sensor, a highly integrated low-power inertial measurement unit that provides acceleration and angular rate measurement with a triaxial system [8]. Despite including these technological improvements, the AX6 remains a cost-effective option compared to other accelerometers with similar features [9].

Given the growing widespread use of Axivity accelerometers in clinical and epidemiological studies for measuring PA, it is crucial to evaluate the reliability of these devices. Although several studies have investigated the inter- and intra-instrument variability of other accelerometers [10,11,12,13,14], the technical variability—both inter- and intra-instrument—of the Axivity AX6 remains unexplored.

Using a vibration platform to assess accelerometer accuracy before human free-living studies offers significant methodological advantages, including generating a wide range of accelerations, recording data from multiple devices simultaneously, and ensuring reproducible oscillations across trials [15]. However, confirming the device’s performance in real human motion remains essential, as demonstrated in prior investigations [2,14].

Thus, the aim of this study was to assess the Axivity AX6 accelerometer’s intra- and inter-instrument variability and reliability across a range of accelerations representative of free-living PA, tested under both technical conditions (using a vibration platform) and human motion conditions (walking and running).

## 2. Materials and Methods

### 2.1. Accelerometer Device

The Axivity AX6 (Axivity Ltd., Newcastle upon Tyne, UK) monitor is a cost-efficient accessible wearable device designed to objectively measure physical movements. It features a Bosch BMI160 sensor, which includes a configurable triaxial accelerometer (±2–16 g) and a triaxial gyroscope (125–2000°/s), both of which can be configured with variable sampling rates (10, 30 or 100 Hz) via Open Movement software (OMGUI, version 1.0.0.43; Newcastle University, UK) [16]. The device is lightweight (11 g), compact (dimensions of 23 × 32.5 × 8.9 mm), and is equipped with a 1024 MB memory and a rechargeable lithium battery (capacity of 250–280 mAh).

The AX6 captures motion data along three axes: horizontal right–left (*X*), vertical (*Y*), and horizontal front–back axis (*Z*), using a solid-state triaxial accelerometer (Figure 1).

A global value measuring the signal vector magnitude (SVM) can be derived from the three accelerometer axes using the OMGUI software [16]. The SVM represents the vector magnitude of the accelerometer data, calculated as the Euclidean distance of the three axes, with 1 g due to gravity subtracted when the device is stationary $\left(\sqrt{x^{2}+y^{2}+z^{2}}-1\right)$ [17]. This calculation can be performed with or without the application of a 4th order Butterworth band-pass filter, after computing the Euclidean norm, with ω_0_ = 0.5–20 Hz [17]. Finally, after applying the band-pass filter, negative values can be dealt with by either taking the ‘absolute’ value (default) or clipping them to zero.

Although the OMGUI software typically reports data in gravity (g), where 1 g corresponds to 9.81 m·s^−2^, for our study we converted these values to milligravity (mg) because mg is more commonly used in physical activity research, facilitating comparison across studies. SVM in mg provides a quantitative measure of PA, with higher values corresponding to greater PA intensity during the measurement period.

### 2.2. Experimental Design

This study comprised two experiments to assess the Axivity AX6 accelerometer’s performance. The first experiment evaluated technical variability and reliability using a vibration platform, and the second examined device performance during human motion scenarios. To record accelerations, AX6 devices were programmed with OMGUI software at a sample frequency of 100 Hz and a dynamic range of ±8 g [16], in line with standard procedures for measuring daily PA in humans [18,19]. We selected 60 s in the Epoch section of the OMGUI software to obtain 60 data/min. The signal was improved by filtering for only frequencies of human motion (high-frequency noise), by applying a 4th order Butterworth band-pass filter. Therefore, a band-pass filter was applied, followed by the Euclidean norm [20]. Negative values were dealt with by clipping them to zero ($max\;(0,\sqrt{x^{2}+y^{2}+z^{2}}-1)$) [17].

In both experiments, the accelerometers were tested under four progressively increasing acceleration conditions. Full details of each experiment are provided below.

#### 2.2.1. Experiment 1: Technical Conditions

We conducted a cross-sectional study in which 12 out of 22 new Axivity AX6 devices were randomly selected. Each device was securely mounted on a vibration platform to prevent unintended movements or accelerometer misalignment. The table was driven by a three-phase motor (JL 712-2; Lafert Electric Motors; Hallam, VIC, Australia), precisely controlled by a compact inverter type (FRNO.75C1S, FRENIC-Mini Series; Fuji Electric FA Components & Systems Co., Ltd., Tokyo, Japan) (Figure 2). To ensure exclusively vertical vibration, each AX6 device was aligned along the *Y*-axis of the vibration tool.

The vibration frequencies used were selected to correspond with the cut-off points defined in the literature in healthy adults for sedentary, light, moderate, and vigorous activity levels [21,22,23]. These levels were replicated by setting the platform to four distinct frequencies while keeping vibration amplitude constant (30 mm). Table 1 summarizes the frequency and corresponding acceleration for each condition. The four testing conditions were administered in random order to minimize the potential order effect.

Each testing condition was maintained for 7 min, with the first and last 1 min excluded from the analysis. This procedure was repeated with the AX6 devices positioned to ensure that vibration occurred exclusively along the *X* and *Z* axes, respectively (Figure 3). The vibration platform was operated for 5 min at each of the four frequencies prior to each test session, as described in previous studies [11,12]. To generate the required accelerations, we employed a vertical shaker with a frequency range of 132–564 (±2) rpm, which thus allowed for a reasonable simulation of repetitive human motions such as gait [10,12,24].

#### 2.2.2. Experiment 2: Human Motion Conditions

In this experiment, 4 out of 22 newly acquired AX6 devices were randomly selected. To assess their performance during human motion, the accelerometers were tested under 4 conditions representative of common walking and running tasks, with progressively increasing speeds [2,24]: (a) 4 km·h^−1^, (b) 6 km·h^−1^, (c) 8 km·h^−1^, and (d) 10 km·h^−1^.

The experiment was conducted on an electronically controlled treadmill (h/p/cosmos, Nussdorf–Traunstein, Germany). Each condition lasted 12 min, with a minimum 10 min break between conditions. The first and last minute of each 12 min bout were excluded, yielding 10 min of valid data per condition.

All 4 monitors were tested simultaneously on a single subject (male, 24 years old; height: 178.8 cm; weight: 70.5 kg; body mass index: 22.1; right-handed), who regularly engaged in physical exercise and had no balance or ambulation disorders. This experiment was conducted in accordance with the Declaration of Helsinki and complied with ethical standards for sport and exercise science research [25].

Each accelerometer was securely placed within an individual wristband. The wristbands were then distributed between the wrists, with two positioned on the left wrist (N1 and N2) and two on the right wrist (N3 and N4), ensuring proper placement and stability (Figure 4).

### 2.3. Statistical Analysis

Statistical analysis was performed using Jamovi software (version 2.4.7) and R software (version 4.3.2). Data are presented as mean ± standard deviation (SD), unless otherwise specified.

In experiment 1, two-way repeated-measures ANOVA was used to assess the effects of frequency (2.2, 3.2, 6.5, and 9.4 Hz) and axis (*X*, *Y*, and *Z*) on accelerometer output. The Greenhouse–Geisser correction was applied when the assumption of sphericity was violated, following established methodologies [11,12]. Intra-instrument reliability was assessed by calculating the coefficient of variation (CVintra) for each device, based on 5 min time fractions (1–6 min) for each condition, with the coefficient of variation defined as the ratio between the SD and the mean of the data. Inter-instrument reliability was evaluated using intraclass correlation coefficients (ICC) with a two-way random-effects model for absolute agreement, assessing agreement between the accelerometers for each axis (*X*, *Y*, and *Z*) individually and in combination. An ICC close to 1 indicates high repeatability [26]. Additionally, the coefficient of variation between instruments (CVinter) was calculated for each axis at each frequency.

A two-way repeated-measure ANOVA test was again performed to examine the effects of device (4 AX6 units) and human motion condition (walking/running at 4, 6, 8, and 10 km·h^−1^) on accelerometer output (experiment 2). To assess the relevance of the observed variations in the Axivity AX6 accelerometers under human motion conditions, the minimum detectable change (MDC) was calculated using the formula $MDC=(SDpooled\;\cdot \;\sqrt{1-ICC})\;\cdot \;(1.96\cdot \;\sqrt{2})$, where $SDpooled=\sqrt{\frac{{SDNx}^{2}{+SDNy}^{2}}{2}}$ (Nx and Ny refer to 2 different AX6 devices) [27,28]. The ICC was calculated for absolute concordance in a mixed-effects model using the acceleration values of each device in each motion condition.

CVintra was also calculated for each device to assess intra-instrument reliability, based on 10 min time fractions (1–11 min) for each motion condition. Finally, inter-instrument reliability was assessed using ICC to evaluate agreement between each accelerometer individually and in combination, and the CVinter was calculated for each device and human motion condition.

## 3. Results

### 3.1. Experiment 1: Technical Conditions

#### 3.1.1. Frequency and Axis Effects

Acceleration output (mg) increased across all axes with increasing frequency (Table 2), with the differences being statistically significant (all *p* < 0.001). However, the two-way ANOVA test for acceleration revealed no significant main effect for axis and no significant frequency*axis interaction effect (*p* = 0.712 and *p* = 0.336, respectively). Post hoc tests indicated no significant differences (*p* > 0.05) between any of the three axes.

#### 3.1.2. Intra-Instrument Variability

The CVintra values for each axis and frequency are shown in Table 3. The average CVintra values across all axes and frequencies ranged from 3.3% to 5.5%.

#### 3.1.3. Inter-Instrument Variability

The ICC for acceleration across all frequencies, combining the three axes, was 0.98. The ICC values for the *X*, *Y*, and *Z* axes individually were 0.98, 0.98, and 0.97, respectively. CVinter took the highest values at 9.4 Hz, considering all three axes. The CVinter values were similar across all other frequencies (2.2, 3.2, and 6.5 Hz) and axes (Table 4).

### 3.2. Experiment 2: Human Motion Conditions

#### 3.2.1. Condition and Device Effects

Acceleration output (mg) increased for all devices as activity intensity increased (Table 5), with differences being statistically significant (all *p* < 0.001). The two-way ANOVA indicated a significant main effect of the device and a significant interaction effect conditions*device (*p* < 0.001). Post hoc analysis (Tukey test) further confirmed significant differences among the four devices (*p* < 0.001).

The mean MDC was calculated for each condition: walking at 4 km·h^−1^: 15.7 mg; walking at 6 km·h^−1^: 22.1 mg; running at 8 km·h^−1^: 80.4 mg; running at 10 km·h^−1^: 94.6 mg. The observed range of mean inter-device difference was lower than MDC in all four human motion conditions, supporting that the inter-device difference was not meaningful in the practical sense (walking at 4 km·h^−1^: 2.9–12.2 mg; walking at 6 km·h^−1^: 5.4–18.9 mg; running at 8 km·h^−1^: 18.0–74.4 mg; running at 10 km·h^−1^: 28.1–89.7 mg).

#### 3.2.2. Intra- and Inter-Instrument Variability

The CVintra values for each axis and frequency are shown in Table 6. Average CVintra values for all motion conditions ranged between 5.3% and 8.8%.

The ICC across all human conditions, combining the four devices, was 0.98. When analyzed separately by human condition, the ICC values for each of the four accelerometers were consistently 0.99. The highest CVinter was observed during walking at 4 km·h^−1^ (10.4%), considering all four devices (Table 6). However, the values were similar for the other three motion conditions (7.1–9.4%).

## 4. Discussion

According to our results, the Axivity AX6 accelerometer demonstrated high intra- and inter-instrument technical reliability across an acceleration range (from 30 mg to 520 mg) commonly proposed for classifying different levels of PA [21,22,23]. Similarly, the AX6 also revealed good intra- and inter-instrument human motion reliability in human motion conditions. Although further studies are needed to evaluate the device’s reliability in free-living conditions, our findings provide preliminary evidence supporting its measurement consistency.

The combined ICC across all conditions (2.2, 3.2, 6.5, and 9.4 Hz) and axes (*X*, *Y*, and *Z*) was 0.98, indicating strong agreement between devices [25]. This aligns with findings reported in other studies that tested different accelerometer brands, such as the Actigraph GT3X (ICC = 0.97) [12], RT3 (ICC = 0.99) [11], Actigraph 7164 (ICC = 0.995) [13], Actical (ICC = 0.985), and Vivago (ICC = 0.89) [13,14]. Considering each individual axis, our results (ICC_X_ = 0.98; ICC_Y_ = 0.98; ICC_Z_ = 0.97) showed a good intraclass correlation coefficient between the measurements, in accordance with other studies that tested GT3X (ICC_X_ = 0.99; ICC_Y_ = 0.98; ICC_Z_ = 0.98) [12] and RT3 (ICC_X_ = 0.99; ICC_Y_ = 0.99; ICC_Z_ = 0.99) [11] accelerometers. In the human motion experiment, the combined ICC for the four AX6 devices was 0.98, consistent with the technical experiment. Similar values were observed when analyzing individual devices (ICC = 0.99 for all), reinforcing the strong agreement between devices under the technical experiment, as well as coinciding with the results of similar studies with other accelerometers [2].

In our work, no significant differences were observed between the axes under any of the four technical testing conditions (Table 2). However, other studies that reported the absolute values for axis-specific activity or acceleration have found differences at high frequencies [11,12]. This indicates that the Axivity AX6 is more reliable under high technical acceleration conditions compared to other different devices such as the RT3 or GT3X [11,12], as the Axivity AX6 renders very similar measurement values between its three axes. This is of high interest for research studies that use wearables to monitor daily PA, based on the importance of providing reliable measurements in all motion conditions.

However, when comparing four AX6 accelerometers during human motion conditions, significant differences were observed in the mean acceleration value across conditions and devices (*p* < 0.001, Table 5). This could be explained by the tangential acceleration (tangencial acceleration=mean angular acceleration·radius), where devices placed more distally on the arm (N1 and N3) recorded higher acceleration that those positioned proximally (N2 and N4) [29]. Accelerations recorded on the left (N1 and N2) and right (N3 and N4) wrists also differed significantly (*p* < 0.001) from each other, due to the asymmetry of motion even in periodic activities. The differences between devices may also be explained by the asymmetric nature of upper limb motion, even during cyclic tasks such as walking or running. Specifically, devices N1 and N2 were placed on the left wrist, and N3 and N4 on the right wrist (Figure 4). Prior research has shown that interlimb asymmetries are inherent to human motion, including rhythmic gait patterns, leading to differing acceleration profiles between dominant and non-dominant limbs [29]. Therefore, a portion of the inter-device variability observed in our study can be attributed to this natural asymmetry. Although our experimental design allowed us to assess the reproducibility of the devices under real motion conditions, future studies aiming to compare device equivalence should consider placing all sensors on the same limb or body location to minimize variability induced by motor asymmetries. Despite these differences, it is important to note that they did not exceed the MDC, indicating that while variations exist, they do not affect the interpretation of PA level or the practical implications of accelerations measurements.

The Axivity AX6 exhibited low CVintra and CVinter across all tested frequencies in the technical experiment (Table 3 and Table 4), and for all devices in each condition in the human motion experiment (Table 6). The mean CVintra for all axes and frequencies ranged from 3.3% to 5.5% (Table 3), indicating very low CVintra. These results are in agreement with those reported in other technical studies for the Actigraph 7164 (4.1%) [13], Actical (0.4%) [13], Vivago (10.9%), and RT3 (4.3%) [11,14]. In the GT3X accelerometer technical variability study [12], values similar to ours (0.4–2.5%) were reported for intermediate frequencies (2.1, 3.1, 4.1 Hz). However, for very low and very high frequencies (1.1 and 10.2 Hz), the mean CVintra for the *Y*-axis was considerably higher, with values of 18.5% at 1.1 Hz and 27.3% at 10.2 Hz.

Overall, the Axivity AX6 showed a CVinter of 6.8%, indicating low deviation from the mean across the 12 devices analyzed in technical conditions and, thus, minimal inter-device variability. In our study, the values of CVinter for the three lowest frequencies (2.2, 3.2, and 6.5 Hz) were relatively low across all axes (3.9–5.7%). In contrast, at the highest frequency (9.4 Hz), the CVinter increased to 14.7% for the *Z*-axis. Esliger et al. [13] found a mean CVinter of 4.9% for the Actigraph 7164, which closely aligns with our results and those of Vanhelst et al. [14] for the Vivago accelerometer (8.9%). Powel et al. [11] also reported CVinter for the RT3 by axes and conditions, with minimal differences between axes, as observed in our study. However, their values were higher for low (2.1 Hz) frequencies (21.9–26.7%) and lower for medium (5.1 Hz) and high (10.2 Hz) frequencies (6.3–9.0% and 4.2–7.2%, respectively). Santos-Lozano et al. [12] reported wide variations in CVinter across axes and frequencies for the GT3X accelerometer, with the lowest values (<10%) at 2.1–4.1 Hz. However, there was high inter-instrument variation at the lowest tested frequency (1.1 Hz), with CVinter values > 149%, and at the highest tested frequency (10.2 Hz), with values ranging between 52.6% and 99.5%. The results obtained in human motion are in line with those described in the technical analysis, with low mean values for both CVintra (5.3–8.8%) and CVinter (7.1–10.4%). In the same way, the results of the second experiment are consistent with the data provided by other studies similar to this one [2,14].

The technical and human motion intra- and inter-device variation between AX6 devices was generally comparable to or smaller than the variations reported for other devices in previous studies, indicating that the acceleration measurements collected by the AX6s were consistent. This confirms that the results remain highly similar across devices and motion conditions. Additionally, although some differences between devices were observed, they did not significantly impact the interpretation of PA level.

Our findings provide preliminary evidence of the technical and human motion variability of the Axivity AX6 accelerometer. However, further research is required to assess its reliability in free-living conditions, where unstructured motion, varying environmental factors, and potential device displacement may introduce additional sources of variability not accounted for in our controlled experiments.

## Figures and Tables

**Figure 1 sensors-25-02480-f001:**
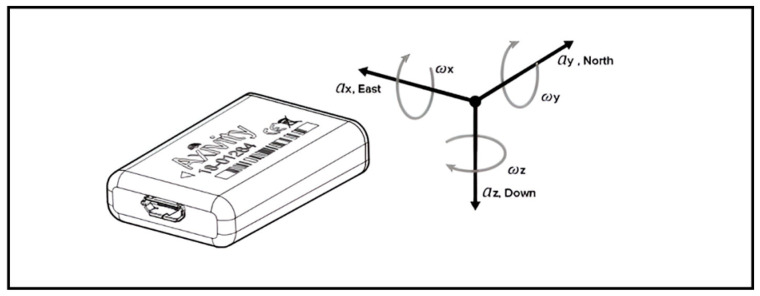
Axivity AX6 device and its three axes of motion.

**Figure 2 sensors-25-02480-f002:**
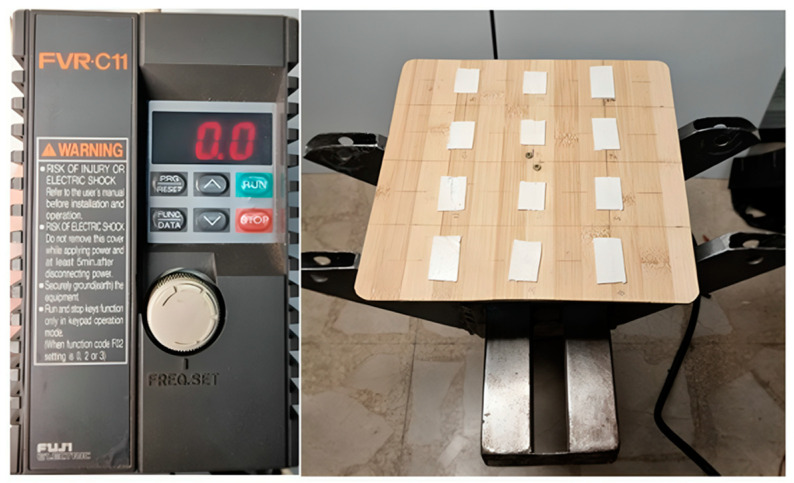
Three-phase motor with compact inverter and vibration platform.

**Figure 3 sensors-25-02480-f003:**
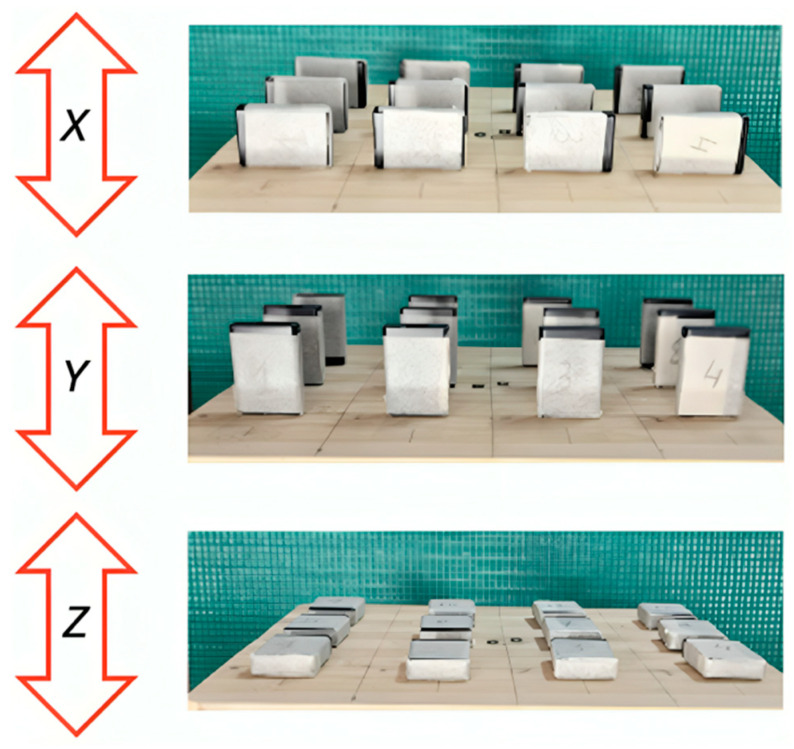
Placement of the accelerometers on the vibration platform to be shaken vertically in the *X*, *Y*, and *Z* axes. The arrow indicates the direction of motion.

**Figure 4 sensors-25-02480-f004:**
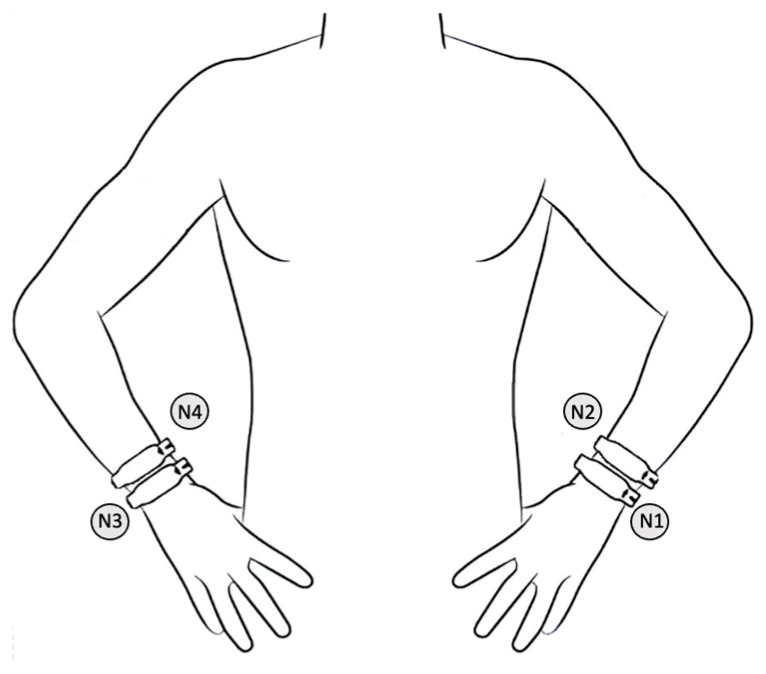
Accelerometer placement and wrist location.

**Table 1 sensors-25-02480-t001:** Description of the four testing conditions used.

Condition	Frequency (Hz) *	Acceleration (mg)
Sedentarism	2.2	~30
Light	3.2	~53
Moderate	6.5	~255
Vigorous	9.4	~520

Hz: hertz/mg: milligravity. * The amplitude was kept at 30 mm for all frequencies.

**Table 2 sensors-25-02480-t002:** Acceleration (mg) at each axis and frequency.

Frequency (Hz)	Axes		
	*X* (mg)	*Y* (mg)	*Z* (mg)
2.2	28.1 ± 1.3	28.7 ± 1.4	30.7 ± 1.6
3.2	59.1 ± 2.8 ^a^	61.7 ± 2.5 ^a^	62.9 ± 3.5 ^a^
6.5	258.2 ± 9.9 ^a, b^	269.1 ± 12.3 ^a, b^	253.0 ± 14.5 ^a, b^
9.4	554.5 ± 68.6 ^a, b, c^	517.3 ± 61.45 ^a, b, c^	534.5 ± 78.3 ^a, b, c^

Data are mean ± standard deviation. ^a^ *p* < 0.001 vs. 2.2 Hz at the same axis. ^b^ *p* < 0.001 vs. 3.2 Hz at the same axis. ^c^ *p* < 0.001 vs. 6.5 Hz at the same axis.

**Table 3 sensors-25-02480-t003:** Intra-instrument coefficient of variation (CVintra) for the mean acceleration recorded at each axis and frequency.

Frequency (Hz)	Axes		
	*X* (%)	*Y* (%)	*Z* (%)
2.2	3.7 (3.4–4.3)	4.4 (3.7–5.4)	4.1 (3.5–4.7)
3.2	3.4 (2.8–4.3)	3.5 (3.3–3.8)	3.7 (2.9–4.9)
6.5	3.6 (2.7–4.4)	3.3 (1.5–5.5)	4.5 (3.0–6.8)
9.4	3.5 (2.4–4.8)	3.4 (2.3–5.6)	5.5 (3.9–8.8)

Data are shown as mean and range (min–max).

**Table 4 sensors-25-02480-t004:** Inter-instrument coefficient of variation (CVinter) for the mean acceleration recorded at each axis and frequency.

Frequency (Hz)	Axes		
	*X* (%)	*Y* (%)	*Z* (%)
2.2	4.4	5.0	5.1
3.2	4.7	4.0	5.5
6.5	3.9	4.6	5.7
9.4	12.4	11.9	14.7
Overall mean	6.3	6.4	7.7

**Table 5 sensors-25-02480-t005:** Acceleration (mg) at each device and human condition.

Conditions		Devices			
		N1	N2	N3	N4
Walking	4 km·h^−1 d^	89.8 ± 8.6	85.2 ± 7.9	80.3 ± 6.5	77.5 ± 6.6
	6 km·h^−1 d^	139.4 ± 11.0 ^a^	132.6 ± 10.5 ^a^	125.8 ± 9.1 ^a^	120.4. ± 9.2 ^a^
Running	8 km·h^−1 d^	624.0 ± 36.0 ^a, b^	595.2 ± 32.6 ^a, b^	576.8 ± 29.6 ^a, b^	549.8 ± 26.6 ^a, b^
	10 km·h^−1 d^	677.0 ± 47.1 ^a, b, c^	648.9 ± 41.9 ^a, b, c^	619.8 ± 39.8 ^a, b, c^	587.3 ± 36.6 ^a, b, c^

Data are mean ± standard deviation. ^a^ *p* < 0.001 vs. 4 km·h^−1^ at the same device. ^b^ *p* < 0.001 vs. 6 km·h^−1^ at the same device. ^c^ *p* < 0.001 vs. 8 km·h^−1^ at the same device. ^d^ *p* < 0.001 among all devices at the same condition.

**Table 6 sensors-25-02480-t006:** Intra- and inter-instrument coefficient of variation for the mean acceleration recorded at each human condition.

Conditions		CVintra (%)	CVinter (%)
Walking	4 km·h^−1^	8.8 (8.1–9.4)	10.4
	6 km·h^−1^	7.7 (7.2–8.0)	9.4
Running	8 km·h^−1^	5.3 (4.9–5.8)	7.1
	10 km·h^−1^	6.5 (6.2–7.0)	8.4

CVintra data are shown as mean and range (min–max).

## Data Availability

Data are contained within the article.

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
