# Peer review of "Variability and Reliability of the Axivity AX6 Accelerometer in Technical and Human Motion Conditions"

_sensors, 2025, doi:10.3390/s25082480_

Round 1

Reviewer 1 Report

Comments and Suggestions for Authors

The subject of the manuscript is technical in its focus, and although the study is not very novel in its questions, methods or results, its results could be certainly useful. Although the authors approach the field from the direction of the study of physical activity for sport and education, the question of the reliability (and comparability) of measurements with different commercial tools is highly relevant in the field of actigraphy and from a medical point of view.

The style of the work is a little too technical report-like, not the best edited, and the feeling of incoherence is reinforced by the fact that there is one sentence in Spanish left in the text.

These would be minor issues in themselves (some of them are highlighted below), but there is also a more serious problem that calls into question the correctness of the results.

Describing the pre-processing steps of the measured acceleration data is an aspect that many studies erroneously neglect, making it difficult to compare results later. The manuscript correctly describes that the 3-axis acceleration data can be pre-processed in several ways: eliminating the gravitational acceleration in several ways, and then computing a magnitude of acceleration. The problem is that the authors choose a flawed combination of methods that unnecessarily distorts the results.

Filtering at lower frequencies using a band-pass filter between 0.5-20 Hz serves precisely the purpose of removing 1 g (or some projection of it in the case of axial data). Subtracting 1 g from the magnitude after that filtering and then discarding negative values (the latter is the ENMO method) gives a signal other than the magnitude of acceleration of the body part. The removal of gravity of Earth can be achieved by either subtracting 1 g from the magnitude OR by high-pass filtering, but with BOTH of these we have subtracted 2 g.  This is not appropriate, especially if we then discard the negative values. The description on the Github page referenced in the manuscript is also misleading, but this process is by no means a proper processing of the data. Correctly pre-processed data should be used to re-run all the statistical tests presented in the article.

For ENMO, and pre-processing techniques of actigraphic acceleration signal in general, I recommend the following two articles:

- van Hees VT, Gorzelniak L, Dean León EC, Eder M, Pias M, Taherian S, et al. (2013) Separating Movement and Gravity Components in an Acceleration Signal and Implications for the Assessment of Human Daily Physical Activity. PLoS ONE 8(4): e61691. https://doi.org/10.1371/journal.pone.0061691

- Maczák B, Vadai G, Dér A, Szendi I, Gingl Z (2021) Detailed analysis and comparison of different activity metrics. PLoS ONE 16(12): e0261718. https://doi.org/10.1371/journal.pone.0261718

In the latter, ENMO is used as an activity metric, but it can also be considered as pre-processing, as the authors have done here.

A further note on the interpretation of the results: when comparing the 4 devices placed on the human body, ANOVA showed a significant difference. The authors only explain that the difference between the N2 and N4 and N1 and N3 devices is due to the formers being placed higher on the wrists than the latters.  

The difference between N1, N2 and N3, N4 is obviously due to the fact that they are positioned on two separate wrists, since all movements are asymmetrical, even in the case of periodic movements. It also does not seem an appropriate choice to compare accelerometers on different parts of the body, since they will measure different motions. Understandably, it is an interesting question to what extent the physical activity of the same person is similarly characterised by the devices on the two different arms, but a more detailed analysis of this would be needed. In contrast, it is not mentioned here when this is one of the main reason for the difference in results of the different devices.

Further minor issues:

- Page 4: if all the information is written in the text (lines 124-130), is it redundant to present it in a table.

- Page 5, lines 159-164: the 2.2-9 Hz frequencies are discussed again, redundantly

-Page 5, line 173: "Data output was SVM per minute for each device." - Does this sentence mean that the sum or the average is calculated for 60-second long epochs? If so, it should be mentioned at the presentation of the pre-processing.

- Page 8: Which SD or SDs were used to calculate the MCSD?

- Page 10: Lines 337-337: Non-English sentence.

Comments on the Quality of English Language

- Page 10: Lines 337-337: Non-English sentence.

Author Response

1. Summary

We are pleased to resubmit our revised manuscript, “Technical Variability of the Axivity AX6 Accelerometer,” for the special issue of Sensors entitled Sensing Technology and Wearables for Physical Activity. We sincerely thank the editor and reviewers for their constructive feedback and valuable insights. Below, we summarize our response to reviewer 1's comments. We believe the revisions have addressed all concerns raised and have enhanced the clarity, scientific rigor, and overall quality of the manuscript. Thank you for your consideration

    Yours sincerely,

    Marcos Echevarría Polo, Faculty of Health and Sports Sciences

    University of Zaragoza, mechevarria@unizar.es, +34 640554122

2. Questions for General Evaluation

Reviewer’s Evaluation

Response and Revisions

Does the introduction provide sufficient background and include all relevant references?

Yes/Can be improved/Must be improved/Not applicable

Each response is explained and detailed in the corresponding point-by-point response letter.

Are all the cited references relevant to the research?

Yes/Can be improved/Must be improved/Not applicable

Is the research design appropriate?

Yes/Can be improved/Must be improved/Not applicable

Are the methods adequately described?

Yes/Can be improved/Must be improved/Not applicable

Are the results clearly presented?

Yes/Can be improved/Must be improved/Not applicable

Are the conclusions supported by the results?

Yes/Can be improved/Must be improved/Not applicable

3. Point-by-point response to Comments and Suggestions for Authors

Comments 1: The subject of the manuscript is technical in its focus, and although the study is not very novel in its questions, methods or results, its results could be certainly useful. Although the authors approach the field from the direction of the study of physical activity for sport and education, the question of the reliability (and comparability) of measurements with different commercial tools is highly relevant in the field of actigraphy and from a medical point of view.

The style of the work is a little too technical report-like, not the best edited, and the feeling of incoherence is reinforced by the fact that there is one sentence in Spanish left in the text.

Response 1: We thank the reviewer for the constructive feedback. While the study may not be highly novel in its methodology, we appreciate the reviewer’s recognition of its relevance in the context of actigraphy and medical applications, especially regarding the reliability and comparability of commercial tools used to assess physical activity. We acknowledge that the manuscript had a somewhat technical report-like tone and that a sentence in Spanish was unintentionally left in the text. In response, we have carefully revised the manuscript to improve its academic style, coherence, and readability throughout. The Spanish sentence has been removed, and the text has been thoroughly proofread.

Comments 2: These would be minor issues in themselves (some of them are highlighted below), but there is also a more serious problem that calls into question the correctness of the results.

Describing the pre-processing steps of the measured acceleration data is an aspect that many studies erroneously neglect, making it difficult to compare results later. The manuscript correctly describes that the 3-axis acceleration data can be pre-processed in several ways: eliminating the gravitational acceleration in several ways, and then computing a magnitude of acceleration. The problem is that the authors choose a flawed combination of methods that unnecessarily distorts the results.

Filtering at lower frequencies using a band-pass filter between 0.5-20 Hz serves precisely the purpose of removing 1 g (or some projection of it in the case of axial data). Subtracting 1 g from the magnitude after that filtering and then discarding negative values (the latter is the ENMO method) gives a signal other than the magnitude of acceleration of the body part. The removal of gravity of Earth can be achieved by either subtracting 1 g from the magnitude OR by high-pass filtering, but with BOTH of these we have subtracted 2 g.  This is not appropriate, especially if we then discard the negative values. The description on the Github page referenced in the manuscript is also misleading, but this process is by no means a proper processing of the data. Correctly pre-processed data should be used to re-run all the statistical tests presented in the article.

For ENMO, and pre-processing techniques of actigraphic acceleration signal in general, I recommend the following two articles:

- van Hees VT, Gorzelniak L, Dean León EC, Eder M, Pias M, Taherian S, et al. (2013) Separating Movement and Gravity Components in an Acceleration Signal and Implications for the Assessment of Human Daily Physical Activity. PLoS ONE 8(4): e61691. https://doi.org/10.1371/journal.pone.0061691

- Maczák B, Vadai G, Dér A, Szendi I, Gingl Z (2021) Detailed analysis and comparison of different activity metrics. PLoS ONE 16(12): e0261718. https://doi.org/10.1371/journal.pone.0261718

In the latter, ENMO is used as an activity metric, but it can also be considered as pre-processing, as the authors have done here.

Response 2: We use a Bandpass-Filtered (BP) followed by Euclidian Norm minus one, similar to the one proposed in the study by van Hees et al. in order to eliminate high frequency noise. The analysis is performed with the OMGUI Open Movement software (V1.0.0.43) and the magnitude vector (the acceleration) reported by applying the BP is the option with which the signal is cleaner and less noise is included in the results compared to other methods. We have explained again and more clearly than in the previous version the pre-processing that has been carried out to obtain the magnitude vector and referenced the paper by van Hees (lines 91-94 and 106-111). It is true that with the previous explanation it was implied that -1 was subtracted 2 times and therefore we subtracted a total of 2g, but this was not the case. With the new explanation we think everything is clearer.

The acceleration results we would obtain by removing the Band Pass filter include too much noise in the signal and distort the real acceleration of the activity.

In our study, we use the default band-pass filter provided by OMGUI, which is a fourth order Butterworth filter with cut-off frequencies of 0.5 Hz and 20 Hz and which we cannot modify. On the other hand, to calculate the magnitude vector the only option is to subtract -1 from the Euclidean norm, as pointed out by the software developers and as we have said before.

(lines 92-95 and 107-111)

Comments 3: A further note on the interpretation of the results: when comparing the 4 devices placed on the human body, ANOVA showed a significant difference. The authors only explain that the difference between the N2 and N4 and N1 and N3 devices is due to the formers being placed higher on the wrists than the latters. The difference between N1, N2 and N3, N4 is obviously due to the fact that they are positioned on two separate wrists, since all movements are asymmetrical, even in the case of periodic movements. It also does not seem an appropriate choice to compare accelerometers on different parts of the body, since they will measure different motions. Understandably, it is an interesting question to what extent the physical activity of the same person is similarly characterised by the devices on the two different arms, but a more detailed analysis of this would be needed. In contrast, it is not mentioned here when this is one of the main reason for the difference in results of the different devices.

Response 3: Thank you for your thoughtful comment. We fully agree that asymmetrical movement between limbs, even during rhythmic tasks like walking or running, can explain the differences observed between devices worn on the left and right wrists. We appreciate this clarification and have now revised the manuscript to explicitly address this aspect. Furthermore, we agree that comparing accelerometers placed on different arms is methodologically debatable if the objective is device equivalence. Our aim, however, was not to validate inter-arm equivalence per se, but to assess whether technical variability between AX6 units could be detected under human motion conditions. To clarify this, we have added a dedicated paragraph in the Discussion and acknowledged this limitation in the revised manuscript. (lines 306-218)

Comments 4: Page 4: if all the information is written in the text (lines 124-130), is it redundant to present it in a table.

Response 4:  Thank you for your comment. While the information from lines 124–130 is indeed included in Table 1, we consider that retaining both formats adds value: the narrative in the text helps contextualize the methodological rationale, while the table offers a clear, at-a-glance summary of the experimental conditions. To improve readability and avoid perceived redundancy, we have slightly condensed the text and emphasized its role in guiding the reader through the rationale for the design choices shown in Table 1.

Comments 5:  Page 5, lines 159-164: the 2.2-9 Hz frequencies are discussed again, redundantly

Response 5: Thank you for your comment. We agree that the frequency range 2.2–9.4 Hz had already been clearly presented earlier. We have revised the sentence to remove the redundant reference while retaining the methodological rationale for using a vertical shaker.

Comments 6: Page 5, line 173: "Data output was SVM per minute for each device." - Does this sentence mean that the sum or the average is calculated for 60-second long epochs? If so, it should be mentioned at the presentation of the pre-processing.

Response 6: Thank you for this observation. Actually, in this sentence it is not completely clear what the data output is. It has been removed as this sentence describes redundant information and is better explained in previous paragraphs. The data output is SVM and is explained in line 89 and 92. On the other hand, it is explained that the analysis is performed in epochs of 60 seconds (lines 107-108).

Comments 7: Page 8: Which SD or SDs were used to calculate the MCSD?

Response 7: We appreciate this comment. The method used to calculate the MCSD has been revised to specify the standard deviations involved (lines 202–206), and the results have been updated accordingly (lines 255–259). These changes, marked in red, reflect Reviewer 2’s input and do not modify the conclusions derived from this measure.

Comments 8: Lines 337-337: Non-English sentence.

Response 8: We thank the reviewer for detecting this oversight and apologize for the error… This has now been corrected and replaced with the appropriate English version in the revised manuscript.

Point 1: Page 10: Lines 337-337: Non-English sentence.

Response Point 1: As we pointed out in “Response 1”, the Spanish phrase has been removed.

Reviewer 2 Report

Comments and Suggestions for Authors

This manuscript reports the intra- and inter-instrument variability and reliability of the Axivity Ax6 accelerometer under controlled technical conditions and human motion scenarios. Authors fill in the knowledge gap – no summary on the technical variability of the sensor. Authors have tested the sensors in different conditions, making their findings robust. Authors found appropriate statistical measures and tests to analyze data and make conclusions. This is an important contribution to the field as these devices are used by researchers who assume this finding is true; this supporting evidence will be helpful.

Page 3, line 91: AX6 outputs acceleration data in milligravity (mg) units – is this the only unit? I used this sensor and retrieved data in g, not mg. Please also refer to axivity.com/userguids/omgui/features, under the section head: “Export Raw CSV”. Accelerometer units are either Gravity (g) or Raw sensor units (1/256).

Page 3, line 92: “the absolute changes in acceleration over time” – the phrase is vague. If mg values represent “absolute changes”, then are there values corresponding to “relative changes”? Also, the quantitative measure is the signal vector magnitude (SVM). I think together with the previous sentence, it could be re-written such as “SVM in milligravity (mg) units provides a quantitative measure of PA, with higher values corresponding to greater PA intensity during the measurement period.”

Page 3, line 103: Can authors specify more on the design of the band-pass filter? Window design, filter order, etc.

Page 4, lines 125-127: Please specify that the listed cut-off points were reported for which population (ex. adult or children, age-range, typical/atypical, etc). This is relevant as different cut-off points exist for different populations, and we have to acknowledge it.

Page 6, lines 198-199: Please elaborate more on the calculation of CVintra. How was this 5-minute recording split? For example, was it separated to five 1-minute bins and acceleration of each bin was calculated? Also, how was acceleration measured? Given the descriptions in previous pages, SVM was derived and further processed by dropping negative values. Are reported acceleration values (ex. table 2) the means of non-zero SVM values? It would be helpful if authors can do a more walk-through with respect to how acceleration value was derived for each sensor under each condition.

Page 6, lines 209-210: Please cite the source of the definition for the minimum clinically significant difference. Also, what is ‘SD’ in the formula? If it is (pooled) standard deviation of measured values, the formula is a simplified form of Minimum Detectable Change (MDC), not Minimum Clinically Important Difference (MCID) – the term more common that what authors use. Different methods of calculating MCID is described here for example: Franceschini et al. (2023), doi.org/10.1177/03635465231152484. In comparison, you can check the formula for MDC here: Lindvall et al., (2024), doi.org/10.1016/j.cccb.2024/100222. Either case, the preceding clause (“To assess the relevance…”) is not accurate. If authors are reporting MDC (which I think this is what authors meant), what I assume is that they wanted to assess if difference in measurements between sensors under human motion conditions (should there be any) is significantly greater than measurement noise. Please consider re-phrasing the rationale of calculating a difference measure.

Page 8, line 250: Please specify which post hoc analysis (ex. Tukey HSD) was performed.

Page 8, lines 260-264:  A difference measure cannot be negative. Please correct. Also, “the MCSD remained below the range of mean inter-device difference” is vague. It should rather be written like: “The observed range of mean inter-device difference was lower than MCSD (again, I argue it is MDC, not MCSD) in all four human motion conditions, supporting that the inter-device difference was not meaningful in the practical sense.”

Page 9, lines 298-299: Please specify that these studies used different accelerometers. It’s confusing.

Page 10, lines 337-339: Delete (or translate to English – but it appears to be repetitive information).

Page 10, lines 350-351. Regarding the statement “However, further research is required to assess its reliability in free-living conditions.” Please expand a bit on what you mean, perhaps give a couple examples. What differences are expected in free-living conditions that need to be considered?

Related Question to point above: You only assessed a sampling rate of 100. Why did you choose this, and can you comment on how your results might differ or not if other sampling rates were selected?

It is unclear why data availability statement is marked as “Not Applicable” (Page 11, lines 364). Please explain.

Author Response

1. Summary

We are pleased to resubmit our revised manuscript, “Technical Variability of the Axivity AX6 Accelerometer,” for the special issue of Sensors entitled Sensing Technology and Wearables for Physical Activity. We sincerely thank the editor and reviewers for their constructive feedback and valuable insights. Below, we summarize our response to reviewer 1's comments. We believe the revisions have addressed all concerns raised and have enhanced the clarity, scientific rigor, and overall quality of the manuscript. Thank you for your consideration

    We look forward to hearing from you.

    Yours sincerely,

    Marcos Echevarría Polo, Faculty of Health and Sports Sciences

    University of Zaragoza, mechevarria@unizar.es, +34 640554122

2. Questions for General Evaluation

Reviewer’s Evaluation

Response and Revisions

Does the introduction provide sufficient background and include all relevant references?

Yes/Can be improved/Must be improved/Not applicable

Each response is explained and detailed in the corresponding point-by-point response letter.

Are all the cited references relevant to the research?

Yes/Can be improved/Must be improved/Not applicable

Is the research design appropriate?

Yes/Can be improved/Must be improved/Not applicable

Are the methods adequately described?

Yes/Can be improved/Must be improved/Not applicable

Are the results clearly presented?

Yes/Can be improved/Must be improved/Not applicable

Are the conclusions supported by the results?

Yes/Can be improved/Must be improved/Not applicable

3. Point-by-point response to Comments and Suggestions for Authors

Comments 1: This manuscript reports the intra- and inter-instrument variability and reliability of the Axivity Ax6 accelerometer under controlled technical conditions and human motion scenarios. Authors fill in the knowledge gap – no summary on the technical variability of the sensor. Authors have tested the sensors in different conditions, making their findings robust. Authors found appropriate statistical measures and tests to analyze data and make conclusions. This is an important contribution to the field as these devices are used by researchers who assume this finding is true; this supporting evidence will be helpful.

Page 3, line 91: AX6 outputs acceleration data in milligravity (mg) units – is this the only unit? I used this sensor and retrieved data in g, not mg. Please also refer to axivity.com/userguids/omgui/features, under the section head: “Export Raw CSV”. Accelerometer units are either Gravity (g) or Raw sensor units (1/256).

Response 1: You are correct in noting that the default software output is in gravity (g). However, in physical activity research, it is more common to report accelerometry data in milli-gravity (mg). We therefore converted the raw data from g to mg to remain consistent with typical practice in this field. We have now provided a clearer explanation of this point in lines 97–99 of the manuscript.

Comments 2: Page 3, line 92: “the absolute changes in acceleration over time” – the phrase is vague. If mg values represent “absolute changes”, then are there values corresponding to “relative changes”? Also, the quantitative measure is the signal vector magnitude (SVM). I think together with the previous sentence, it could be re-written such as “SVM in milligravity (mg) units provides a quantitative measure of PA, with higher values corresponding to greater PA intensity during the measurement period.”

Response 2: Thank you for pointing out the ambiguity regarding “absolute changes in acceleration.” We agree that the phrase is vague and have revised the manuscript text to clarify that we are referring to the signal vector magnitude (SVM) measured in mg, which provides a quantitative measure of physical activity (PA). As you suggest, we have also added a statement indicating that higher SVM values correspond to greater PA intensity (lines 99-100).

Comments 3: Page 3, line 103: Can authors specify more on the design of the band-pass filter? Window design, filter order, etc.

Response 3: We appreciate your comment about the filter design. In our study, we used the default band-pass filter provided by OMGUI, which is a fourth order Butterworth filter with cutoff frequencies of 0.5 Hz and 20 Hz. We have updated the manuscript to clarify these filter specifications. The changes are marked in blue because reviewer 1 also commented on them (line 107-111).

Comments 4: Page 4, lines 125-127: Please specify that the listed cut-off points were reported for which population (ex. adult or children, age-range, typical/atypical, etc). This is relevant as different cut-off points exist for different populations, and we have to acknowledge it.

Response 4: Thank you for highlighting this important point. The cut-off points referenced here were originally reported in studies focusing on healthy adult populations. We acknowledge that different cut-off points may be more appropriate for other age groups or specific populations, and we have clarified this detail in the revised manuscript (line 128).

Comments 5: Page 6, lines 198-199: Please elaborate more on the calculation of CVintra. How was this 5-minute recording split? For example, was it separated to five 1-minute bins and acceleration of each bin was calculated? Also, how was acceleration measured? Given the descriptions in previous pages, SVM was derived and further processed by dropping negative values. Are reported acceleration values (ex. table 2) the means of non-zero SVM values? It would be helpful if authors can do a more walk-through with respect to how acceleration value was derived for each sensor under each condition.

Response 5: Thank you for this query. In the revised manuscript (lines 107–111), we have clarified how the accelerometry data were processed and how the SVM was derived. This also facilitates understanding of the CVintra analysis detailed in lines 191 and 208. Specifically, we discarded the first and last minute of each 7-minute trial, resulting in a 5-minute window with 60 data points per minute (i.e., 300 data points per min). In Table 2, the reported acceleration values (mg) for each axis and frequency are presented as mean ± standard deviation, as indicated in the table note.

Comments 6: Page 6, lines 209-210: Please cite the source of the definition for the minimum clinically significant difference. Also, what is ‘SD’ in the formula? If it is (pooled) standard deviation of measured values, the formula is a simplified form of Minimum Detectable Change (MDC), not Minimum Clinically Important Difference (MCID) – the term more common that what authors use. Different methods of calculating MCID is described here for example: Franceschini et al. (2023), doi.org/10.1177/03635465231152484. In comparison, you can check the formula for MDC here: Lindvall et al., (2024), doi.org/10.1016/j.cccb.2024/100222. Either case, the preceding clause (“To assess the relevance…”) is not accurate. If authors are reporting MDC (which I think this is what authors meant), what I assume is that they wanted to assess if difference in measurements between sensors under human motion conditions (should there be any) is significantly greater than measurement noise. Please consider re-phrasing the rationale of calculating a difference measure.

Response 6: Thank you for this important clarification. We have been reviewing the Lindvall et al. manuscript and we believe that this calculation is more correct for our study. The MDC calculation is explained in lines 202-206 of our manuscript. In comment 10 we discuss about the new MDC results obtained .

Comments 7: The difference between N1, N2 and N3, N4 is obviously due to the fact that they are positioned on two separate wrists, since all movements are asymmetrical, even in the case of periodic movements. It also does not seem an appropriate choice to compare accelerometers on different parts of the body, since they will measure different motions. Understandably, it is an interesting question to what extent the physical activity of the same person is similarly characterised by the devices on the two different arms, but a more detailed analysis of this would be needed. In contrast, it is not mentioned here when this is one of the main reason for the difference in results of the different devices.

Response 7:  Thank you for reiterating this point, which was also raised by Reviewer 1. As you correctly note, exploring differences arising from distinct sensor placements would require a different study design and more extensive analyses, and therefore lies beyond our present scope. Nonetheless, we agree it is important to acknowledge this issue. We have now added a brief note in the Discussion (lines 306–318 of the revised manuscript) stating that accelerometers on different limbs or anatomical locations inherently capture slightly different motion profiles due to natural asymmetry in human movement.

Comments 8: Page 8, line 250: Please specify which post hoc analysis (ex. Tukey HSD) was performed.

Response 8: It was Tukey, now it has been detailed (line 245).

Comments 9:  Page 8, lines 260-264:  A difference measure cannot be negative. Please correct. Also, “the MCSD remained below the range of mean inter-device difference” is vague. It should rather be written like: “The observed range of mean inter-device difference was lower than MCSD (again, I argue it is MDC, not MCSD) in all four human motion conditions, supporting that the inter-device difference was not meaningful in the practical sense.”

Response 9:  Thank you for spotting this detail. We have changed the previous values, because with the new MDC calculation the values were lower, i.e. the margin of error was smaller. However, the conclusions are the same, because the differences between accelerometers do not exceed the values obtained by MDC (lines 255-259). On the other hand, we have removed the symbol “±” since, as you have pointed out, the measurement difference could not be negative.

Comments 10: Page 9, lines 298-299: Please specify that these studies used different accelerometers. It’s confusing.

Response 10: Thank you for pointing this out. We have revised the text to clarify that those studies involved different brands or models of accelerometers, which helps avoid confusion regarding how the inter- and intra-instrument variability results compare to ours (lines 297 and 298).

Comments 11: Page 10, lines 337-339: Delete (or translate to English – but it appears to be repetitive information).

Response 11: We apologize for the oversight. The sentence originally written in Spanish has now been removed.

Comments 12: Page 10, lines 350-351. Regarding the statement “However, further research is required to assess its reliability in free-living conditions.” Please expand a bit on what you mean, perhaps give a couple examples. What differences are expected in free-living conditions that need to be considered?

Response 12: Thank you for suggesting this clarification. By “further research in free-living conditions,” we refer to settings where participants’ activities are not strictly controlled, and the accelerometers may face additional challenges such as device displacement, varied environmental conditions, and unstructured movement. These factors can introduce variability in the signals and may affect the measurement of daily physical activity patterns in ways not captured under the controlled conditions we studied (modifications on lines 356-359).

Comments 13: Related Question to point above: You only assessed a sampling rate of 100. Why did you choose this, and can you comment on how your results might differ or not if other sampling rates were selected?

Response 13: We chose a sampling rate of 100 Hz because it is a commonly recommended setting for capturing typical daily movements with sufficient resolution. This rate provides a detailed signal while keeping file sizes and battery usage manageable. Other sampling rates (e.g., 30 Hz or 50 Hz) could be employed based on different study aims, such as prolonged free-living measurements with limited storage or battery constraints. Although slight variations in signal resolution may occur at lower sampling rates, we do not expect large differences in the overall variability and reliability of the Axivity AX6 for the moderate-to-vigorous range of movements assessed in our study.

Comments 14: It is unclear why data availability statement is marked as “Not Applicable” (Page 11, lines 364). Please explain.

Response 14: We do not want to share data sets from this research at this time until all manuscript publications required for MEP’s PhD degree have been completed.

Round 2

Reviewer 1 Report

Comments and Suggestions for Authors

The authors have made appropriate corrections to the manuscript for most of the comments.

However, there is still a problem with the SVM calculation, but fortunately only in terms of how it is presented in the manuscript:

The bandpass filter naturally, as the name suggests, includes both lowpass and highpass filtering. The authors are correct that the lowpass filter is useful for removing high frequency noise, but highpass filtering removes all DC components, such as g in case of the vector magnitude, or some of its projections in case of axial data. This is why BP filter is also used to remove g. Also in the van Hees article cited by the authors, either ENMO is used (1 g removed) OR HFEN, where bandpass filtering is used but then 1 g is not removed.

Thus the order in the new text (line 110) is still incorrect: "Therefore, bandpass-filtered was applied followed by Euclidean Norm [20].". In this case, g had been subtracted twice.
However, both methods can be used together: if ENMO is applied first (then the signal already centralized around 0), and then the BP filtering happens.

So I checked the description of the OMGUI software used to see which order the software uses, but it does not give a clear answer. The articles cited there also applied either filtering only or 1g subtraction only. So I checked the open source code of the software, and fortunately the following came up:
The code first computes a Euclidean norm, then filters this with a BP filter, and then clips out the negative parts. 
This is an unusual procedure, but fortunately not entirely wrong, as the text would suggest. Thus the results are correct, but the description of the correct order is necessary (rewriting lines 92-95).

Further comment: line 108: "...60-second epochs, obtaining 60 data/min." - using 60 second long epochs means we have 1 data/min.

Author Response

Comments 1: The bandpass filter naturally, as the name suggests, includes both lowpass and highpass filtering. The authors are correct that the lowpass filter is useful for removing high frequency noise, but highpass filtering removes all DC components, such as g in case of the vector magnitude, or some of its projections in case of axial data. This is why BP filter is also used to remove g. Also in the van Hees article cited by the authors, either ENMO is used (1 g removed) OR HFEN, where bandpass filtering is used but then 1 g is not removed.

Thus the order in the new text (line 110) is still incorrect: "Therefore, bandpass-filtered was applied followed by Euclidean Norm [20].". In this case, g had been subtracted twice.

However, both methods can be used together: if ENMO is applied first (then the signal already centralized around 0), and then the BP filtering happens.

So I checked the description of the OMGUI software used to see which order the software uses, but it does not give a clear answer. The articles cited there also applied either filtering only or 1g subtraction only. So I checked the open source code of the software, and fortunately the following came up:

The code first computes a Euclidean norm, then filters this with a BP filter, and then clips out the negative parts.

This is an unusual procedure, but fortunately not entirely wrong, as the text would suggest. Thus the results are correct, but the description of the correct order is necessary (rewriting lines 92-95).

Response 1: Thanks for your feedback and comments on this new revision. A couple of comments have been added on the indicated lines (94-96) to clarify the correct order of the acceleration signal processing.

Comments 2: line 108: "...60-second epochs, obtaining 60 data/min." - using 60 second long epochs means we have 1 data/min.

Response 2: Thank you for pointing this out. In the OMGUI software, if you select in the Epoch section: 60 s (as has been our case), you get a CSV with 60 data per minute. Maybe the software uses the terminology wrongly.

It has been explained differently to clarify the steps followed in the study and to facilitate its replication. (lines 108-109)